# Heterogeneous macroeconomic factors' effects on stocks across sizes, styles, and sectors in the South Korean market

**Chulyoung Cho**[ID]<sup></sup>, **Jinseok Yang**[ID]<sup></sup>, **Beakcheol Jang**[ID]*

Graduate School of Information, Yonsei University, Seoul, South Korea

☯ These authors contributed equally to this work.
* bjang@yonsei.ac.kr

## Abstract

Knowledge of the key macroeconomic variables that influence stock volatility across capital sizes, styles, and sectors can provide clues for investment strategies and policy decisions. We use the GARCH-MIDAS model with feature selection to analyze Korean Benchmark Indices from 2009 to 2022. This study maximizes memory retention through an optimal fractional differentiated price series and uses an adaptive lasso penalty for feature selection. The housing price-sales index and realized volatility were consistently influential across most indices and sectors. The GARCH-MIDAS model, paired with these variables, significantly improves long-term stock volatility forecasts. This study underscores the need to monitor housing prices in South Korea because of their substantial effects on long-term stock volatility.

## Introduction

Over the past two decades, the global economy has faced several tumultuous events, such as the 2008 financial crisis, the 2010 Eurozone Debt Crisis, and the COVID-19 pandemic. These events triggered profound macroeconomic shocks, exemplified by significant fluctuations in the stock market [1, 2]. The aftermath of these crises has spotlighted the pivotal role of macroeconomic indicators—such as interest rates and money supply—in influencing market dynamics, a relationship that has become even more pronounced in the post-pandemic world [3]. Evidence increasingly suggests a tangible link between these indicators and stock market fluctuations, pointing towards a complex interplay that demands deeper investigation [4–6]. However, the focus of existing research has largely been confined to the developed markets, with a significant emphasis on the US stock market. This narrow lens raises questions about the universality of these findings, underscoring a critical gap in our understanding of how different market contexts might influence or alter these established dynamics. In this context, the South Korean market emerges as a particularly compelling subject for in-depth study, given its unique position at the nexus of developed and emerging market characteristics. South Korea's economy and stock market have shown distinctive responses to these global macroeconomic shocks, unlike the often-studied like the US [7]. There are limited studies that focus on

**Data Availability Statement:** All macroeconomic
variables files are available from the DANS Data
Station Social Sciences and Humanities (doi:10.
17026/SS/NHZBHM). All indices data underlying
the results presented in the study are available

from Factset (https://www.factset.com/private/markets).

**Funding:** This study was financially supported by National Research Foundation of Korea (https://www.nrf.re.kr/index) in the form of a grant (RS-2023-00273751) awarded to Beakcheol Jand. The funders had no role in study design, data collection and analysis, decision to publish, or preparation of the manuscript.

**Competing interests:** The authors have declared that no competing interests exist.

**Abbreviations:** EX_RATE, Exchange Rate; F_EMV, Financial Crisis Equity Market Volatility Index; H_SALE, Housing Sales Price Index; K_EPU, Korea Economic Policy Uncertainty Index; NHC, New Housing Construction Index; OVERALL_EMV, Overall Equity Market Volatility Index; RET_SALE, Retail Sale Index; RV, Realized Volatility; UNEMP, Unemployment Rate; US_EPU, U.S Economic Policy Uncertainty Index.

economies that combine the characteristics of both emerging and developed markets. This study attempts to fill this gap by examining the South Korean stock market, a unique blend of emerging and developed market characteristics [8]. The uniqueness is rooted in South Korea's rapid industrialization, technological innovation, and the central role in exports, alongside geopolitical tensions and global economic dependencies [9, 10]. Such a complex economic background makes South Korea an ideal case study for investigating the nuanced effects of macroeconomic variables on stock market dynamics. The main objective of this study is to investigate which factor—among traditional macroeconomic indicators and alternative economic indicators—exerts the most significant influence on stock market volatility. This includes examining how these factors influence stock volatility across different sectors, capital sizes, and investment styles (growth or value). Thereby, this study enriches the understanding of hybrid market dynamics and provide extended insights for volatility, offering valuable guidance to investors and policymakers.

Recently, there has been a growing body of research on the impact on the stock markets using a variety of alternative economic indicators in addition to the traditional macroeconomic factors. For instance, the Economic Policy Uncertainty Index is a measure of national Economic Policy Uncertainty [11]. It is derived from the relative frequency of articles in domestic newspapers and is being actively considered in many studies [12–14]. Hua et al. [15] examined how US uncertainty, as indicated by news-implied volatility, impacted the volatility of the Chinese stock market. They found that investors formulating investment strategies should consider this uncertainty. Moreover, equity market volatility trackers (EMV) [11] linked articles about macroeconomic news and outlook––business, sentiment, and financial crisis––to the VIX index, a measure of expected stock market volatility based on option prices. Currently, geopolitical risk is a crucial factor in stock volatility. The Geopolitical Risk Index, which is updated monthly, measures geopolitical risk through articles about military tensions, sanctions or economic cooperation [16, 17]. Tzeng [18] found that using 20 US macroeconomic variables improves the volatility predictability of 11 Asian stock markets when combined forecasting methods are used. In a notable study, Engle et al. [19] found that including economic fundamentals in the volatility model is beneficial for long-term forecasting. Additionally, they found that these macroeconomic fundamentals are important, even for short-term predictions. However, the existing research, mostly based on each country's representative indices, suggests that the impact of these factors on stock volatility can vary according to stock sizes, styles, or sectors.

An important study on the capital size factor is the cross-section of expected returns [20], which posits that more than 90% of a stock's returns can be attributed to company size. Moreover, earlier research found that small companies can expect higher premiums than large companies [21]. This is primarily because small-cap stocks outperform more easily and experience unexpected growth. Large or medium-sized stocks are influenced more by institutions or foreign investors than small-cap firms. By contrast, the major investors in small-cap stocks are individual investors. Nofsinger [22] studied the influence of public information on investor behavior. He discovered that individual investors trade exclusively on positive news, whereas institutional investors react to both positive and negative news through buying or selling. Following positive economic news, institutions and individuals buy and sell large firms following negative economic news. However, small-cap trading does not seem to be driven by macro-level news. Additionally, macroeconomic indicators, and news sentiment indices can be affected differently in each sector to which each stock belongs [23]. For instance, exchange rates fluctuations have varying impacts on sectors focused on exports and imports, depending on specific thresholds. Identifying the factors that significantly influence stock capital size in volatility forecasting can enhance prediction accuracy and portfolio allocation.

The use of macroeconomic factors presents a data frequency issue, as these variables often have low frequencies, while stock market volatility is high. The mixed data sampling(MIDAS) method, which differentiates between short- and long-term stock market volatility, addresses this by incorporating low-frequency variables into the long-term component, solving the data cycle problem without information loss. We employ multivariate Generalized Autoregressive Conditional Heteroskedasticity (GARCH) MIDAS model combined with the adaptive lasso technique to find the utmost impact macroeconomic factors on stock markets [24–29]. As to the inherent limitations, we are not able to completely remove the potential endogeneity issue from omitted variables. However, since we include the most of the well-known macroeconomic variables as control, the model can minimize the endogeneity issue as much as possible.

Our study contributes threefold to the existing literature. First, we examine the diverse effects across all sectors, capital sizes, and investment styles (growth and value), going beyond traditional analyses that focus on macroeconomic indicators and their impact on national indices or specific sectors. It provides a comprehensive and nuanced cross-sectional analysis that enriches our understanding beyond the current literature.

Second, we build on this foundation by expanding the analytical framework to include a broader range of variables that affect sectors, sizes, and styles. Traditional research is often limited to traditional macroeconomic variables or alternative economic indicators. By integrating a variety of indicators, such as the category-specific EPU, GPRI and EMV, along with the Business Sentiment Index (BSI) and the Economic Sentiment Index (ESI), we provide an updated and detailed view of investor behavior and market conditions. This enhanced approach significantly deepens traditional economic analysis.

Third, this study, which focuses on the South Korean stock market, provides in-depth insights and a sophisticated understanding of countries that have characteristics of both developed and emerging markets and are critical to the global economy. This analysis helps investors develop better-informed strategies while expanding our understanding of the dynamics of such hybrid markets. For investors trying to make sense of these complicated market conditions, the knowledge gained from this study is invaluable. In addition, we evaluate the effectiveness of key variables during recent economic challenges, including the post-COVID-19 downturn, the devaluation of the Korean won, escalating inflation, and rising interest rates. We test the predictive power of our selected variables during a period of significant stock market volatility since the second quarter of 2020. The results demonstrate the robustness of our methodology and provide valuable insights into the current financial market.

The remainder of this paper is organized as follows: Section 2 describes the principles of multivariate GARCH-MIDAS with feature selection. The data are introduced in Section 3. Section 4 presents the empirical findings and Section 5 provides the conclusion.

## Empirical methodology

### The GARCH-MIDAS model with multi-variables

The GARCH-MIDAS model breaks down conditional variance into short- and long-term components [19]. The short-term component is high-frequency mean-reverting and has only a temporary effect on overall volatility. The long-term component is measured by historical realized volatility or macroeconomic variables weighted by the MIDAS polynomial and beta-weighting scheme. First, the daily fractional-differenced stock returns a high frequency $r_{i,t}$ on day i = 1,. . .., $N_t$ and in month $t = 1, \ldots, T$, and is denoted as follows:

$$r_{i,t} - E_{i-1,t}(r_{i,t}) = \sqrt{g_{i,t}\tau_t}\varepsilon_{i,t}$$
$$\varepsilon_{i,t}|\Gamma_{i-1,t} \sim N(0,1), \tag{1}$$

where $E_{i-1,t}(\cdot)$ represents the conditional expectation given information $\Gamma_{i-1,t}$ on day $(i-1)$ based the frequency of the exogenous variables. It contains the short-term component $g_{i,t}$ and the long-term component $\tau_t$. The daily expected returns are fixed constant $\mu$. $N_0 = \sum_{t=1}^{T} N_t$ denotes the cumulative count of daily observations. The short-term volatility component accounts for a mean-reverting daily simple GARCH(1,1) process, whereby

$$g_{i,t} = (1 - \alpha - \beta) + \alpha \frac{\left(r_{i-1,t} - \mu\right)^2}{\tau_t} + \beta g_{i-1,t} \tag{2}$$

with constraints of $\alpha > 0$, $\beta > 0$ and $\alpha + \beta < 1$. The long-term volatility component $\tau_t$ with a one-sided exogenous variable filter $X_t$ is denoted by

$$\log(\tau_t) = m + \theta \sum_{k=1}^{K} \varphi_k(\omega_1, \omega_2) X_{t-k} \tag{3}$$

where $\theta$ expresses the effects of macroeconomic variables on long-term volatility. The logarithm of $\tau_t$ maintains positive long-term volatility. With regard to the K lagged observed variable $X_t$, the beta-weighting is applied to each lagged value [24]:

$$\varphi_k(\omega_1, \omega_2) = \frac{(k/(K))^{\omega_1 - 1} \cdot (1 - k(K))^{\omega_2 - 1}}{\sum_{l=1}^{K} (l/(K))^{\omega_1 - 1} \cdot (1 - l(K))^{\omega_2 - 1}} \tag{4}$$

The parameters $\omega_1$ and $\omega_2$ determine the weights $\varphi_k$, and $\varphi_k \geq 0$ for k = 1, . . ., K, and $\sum_{k=1}^{K} \varphi_k = 1$. Beta-weighting schemes can be used to produce decaying, hump-shaped, or U-shaped weights [24]. The beta-weighting allows for equally (e.g., $\omega_1 = \omega_2 = 1$), increasingly (e.g., $\omega_1 > \omega_2$, more weighted for farther observations) or decreasingly weighting (e.g., $\omega_1 < \omega_2$, weighted more for closer observations) schemes, on condition that $\omega_n \geq 1$, with n = 1, 2. In the empirical analysis, profiling the likelihood yields an estimate of the number of delays K. By maximizing the subsequent log-likelihood, the estimated values of the unknown parameters are calculated as follows:

$$\text{LLF} = -\frac{1}{2} \sum_{t=1}^{T} \left[ \sum_{i=1}^{N_t} \left[ \log(2\pi) + \log(g_{i,t}\tau_t) + \frac{(r_{i,t} - \mu)^2}{g_{i,t}\tau_t} \right] \right] \tag{5}$$

The GARCH-MIDAS model has been actively employed in the exploration of relationships between stock volatility and macroeconomic variables, but most of these studies have focused on a single variable [25]. However, several factors influence real stock volatility. Therefore, it is necessary to introduce all potential factors in a single model. In the empirical analysis, we use a diverse range of macroeconomic variables in the long-term equations. The equation for long-term volatility with J exogenous variables is as follows:

$$\log(\tau_t) = m + \sum_{j=1}^{J} \theta_j \sum_{k=1}^{K} \varphi_k(\omega_1, \omega_2) X_{t-k,j} \tag{6}$$

where j is the total number of explanatory variables, and $\theta_j$ is the indicator of how each variable affects the long-term component. The log-likelihood function for the multivariate GARCH--MIDAS with $\Phi$ estimated for all parameters is as follows:

$$\text{LLF}(\Phi) = -\frac{1}{2} \sum_{t=1}^{T} \sum_{i=1}^{N_t} \left[ \log(2\pi) + \log(g_{i,t}(\Phi)\tau_t(\Phi)) + \frac{(r_{i,t} - \mu)^2}{g_{i,t}(\Phi)\tau_t(\Phi)} \right] \tag{7}$$

The greater the number of macroeconomic variables, the larger the number of parameters. However, it is difficult to identify a close effect using a large number of parameters. To address this issue, a feature selection strategy is investigated. Boriss [26] proposed MIDASSO approach to combine the unrestricted MIxed-frequency DAta-Sampling approach (U-MIDAS), and LASSO penalized [27] "elastic net" regression [28]. In this study, we adopt the adaptive lasso regression method for the log-likelihood function with a penalized term for feature selection.

$$\text{PLLF}(\Phi) = -\frac{1}{2}\sum_{t=1}^{T}\sum_{i=1}^{N_t}\left[\log(2\pi) + \log\big(g_{i,t}(\Phi)\tau_t(\Phi)\big) + \frac{(r_{i,t} - \mu)^2}{g_{i,t}(\Phi)\tau_t(\Phi)}\right] - \lambda\sum_{j=1}^{J}\widehat{\omega}_j|\theta_j| \quad (8)$$

It is essential to find the optimal parameter values for $\Phi_\lambda$ by using the given hyper-parameter $\lambda$ to maximize the penalized log-likelihood function with the constraints of $\alpha > 0$, $\beta > 0$ and $\alpha + \beta < 1$ in the high-dimensional data analysis. The adaptive weight is calculated by using $\widehat{\omega}_j = 1/|\widehat{\theta}|^\eta$. According to the findings in the simulation by Zou [29], the adaptive lasso is superior to the traditional lasso technique. The lasso has an approximately 50% probability of missing a true model. However, the adaptive lasso is consistent with feature selection. As the number of data points increases and the error variance decreases, the feature selection problem is expected to become easier. Therefore, we take $\eta = 2$ based on the simulation results [29]. Conventionally, cross-validation and information criteria, including the Akaike information criterion (AIC) [30] and Bayes information criterion [31], are used in model selection. In this study, the optimal parameter $\lambda$ is measured using the Generalized Information Criterion (GIC) introduced by Fan and Tang [32]. By choosing the optimal $\lambda$, we are able to select the strongest impact variables from the estimated optimal parameter values $\Phi_\lambda$. The GIC comprises two parts, that is, measuring the model fitting and penalized model complexity.

$$GIC_\lambda = \frac{1}{N_0}\left\{2[LLF(\Phi) - PLLF_\lambda(\Phi_\lambda)] + a(N_0, p)|\widehat{\theta}_\lambda|\right\}$$
$$a(N_0, p) = \log\{\log(N_0)\} \cdot \log(p) \tag{9}$$

where $N_0$ denotes the total observations and $p = 3J + 1$ in Eq (6) for the long-term component. The quantity $|\widehat{\theta}_\lambda|$ represents the count of non-zero elements in $\widehat{\theta}_\lambda$, where $\widehat{\theta}_\lambda$ is calculated from Eq (9) for a specific $\lambda$ [32]. To obtain the minimum $GIC_\lambda$, we considered a range of $\lambda$ from 0 to $\lambda_{max}$. As the $\lambda$ increases, some parameter $\theta_j$ is contracted to zero and the related variables are excluded. In this manner, we can take the strongest explanatory variables; however, this may cause identification issues. Given that, the beta-weighting parameter $\omega_1$, $\omega_2$ are not identified when those are excluded from the penalized log-likelihood function. To avoid this, we employ an estimation method using polynomial parameter profiling [33]. There are two parts to be estimated: $\Phi_1 = (\mu, \alpha, \beta, m, \theta_1, \theta_2, \ldots, \theta_{18})$ is the main part to show the explanatory power of each macroeconomic variable, whereas the set $\Phi_2 = (\omega_{1,1}, \omega_{1,2}, \omega_{2,1}, \omega_{2,2}, \ldots, \omega_{18,1}, \omega_{18,2})$ that is used for beta-weighting parameters has minimal effect on the result. Hence, we fix the set $\Phi_2 = \widehat{\Phi}_2$ to avoid identification problems.Text.

## Data

### Stock data

Stock price data were sourced from FactSet, a prominent financial data vendor. The in-sample analysis estimation period runs from January 2009 to September 2022, while the out-of-sample comparison period is from October 2022 to December 2022. We selected a comprehensive set of S&P Korea benchmark indices developed by the S&P. The S&P Global BMI is a rule-based

**Table 1.  Index list to analyze size, style, and sector effects.**

| Usage | Index Name |
|---|---|
| Size Effects Analysis | S&P South Korea BMI LargeMidCap TR |
|  | S&P South Korea BMI SmallCap TR |
| Style Effect Analysis | S&P South Korea BMI Growth TR |
|  | S&P South Korea BMI Value TR |
| Sector Effect Analysis | S&P South Korea BMI Communication Services |
|  | S&P South Korea BMI Consumer Discretionary |
|  | S&P South Korea BMI Consumer Staple |
|  | S&P South Korea BMI Energy |
|  | S&P South Korea BMI Financials |
|  | S&P South Korea BMI Healthcare |
|  | S&P South Korea BMI Industrials |
|  | S&P South Korea BMI Information Technology |
|  | S&P South Korea BMI Materials |
|  | S&P South Korea BMI Real Estate |
|  | S&P South Korea BMI Utilities |

index that gauges the performance of global stock markets. This index includes all publicly traded stocks that meet minimum liquidity requirements and boast market values of at least USD $100 million. Consequently, it enables more accurate analyses of the exogenous effects by eliminating noise from individual stocks. The chosen indices are listed in Table 1 and organized according to their usage in the analysis. To examine the actual return rate of an investment or pool of assets during the specified evaluation period, we select indices determined by total returns. The total return typically provides an effective measure of an investment's overall performance, enabling more accurate analytical results. We employ three analysis categories to examine the heterogeneous effects. First, we use 11 sector indices categorized by the Global Industry Classification Standard to analyze sector effects. Size effects are evaluated using indices across the capital range. The S&P South Korea BMI LargeMidCap TR is designed to track the top 85% of the float-adjusted market cap in Korea, whereas the S&P South Korea BMI SmallCap TR tracks the bottom 15% of the market cap, and the assignments for large/mid/small and capital ranges are exhaustive and mutually exclusive [34]. Finally, style (growth/value) effects are measured using the style benchmark indices. Style indices define each stock in the universe as having 100% growth, 100% value, or a mixture of several variables. The growth variables are historical earnings per share growth rate and sales per share growth rate calculated from a linear regression trend, and the average internal growth rate ((ROE)×(1--PayoutRatio)) for five-year historical values. The value variables are shales/cash flow per share to price, dividend yield, and book value from the balance sheet per share to price [35].

## Optimal fractional differentiated price series with maximum memory preservation

Most studies use the ratio of the previous day's closing price to the same day's closing price, or log return, to calculate stock price changes, making the series stationary but losing all memory from the original series [36]. To forecast and capture its characteristics, memory is needed to gauge the price process's deviation from expected long-term volatility. Thus, it is vital to retain as much memory as possible, while ensuring that the price series becomes stationary. This study employs the Fixed-Width Window Fracdiff (FFD) method [37] and a real (non-integer) positive d preserves memory by the following binomial series expansion ω and the observed

**Table 2. Calculated results of optimal d for FFD.**

|  | Optimal d | p-value | Correlation Coefficient |
|---|---|---|---|
| **Growth** | 0.7 | 0.028 | 0.788 |
| **Value** | 0.3 | 0.258 | 0.98 |
| **LargeMidCap** | 0.2 | 0.049 | 0.993 |
| **SmallCap** | 0.4 | 0.038 | 0.939 |
| **Communication Services** | 0.8 | 0.01 | 0.668 |
| **Consumer Discretionary** | 0.1 | 0.013 | 0.998 |
| **Consumer Staple** | 0.6 | 0.017 | 0.87 |
| **Energy** | 0.1 | 0.046 | 0.998 |
| **Financials** | 0.1 | 0.007 | 0.997 |
| **Healthcare** | 0.7 | 0.046 | 0.767 |
| **Industrials** | 0.3 | 0.011 | 0.965 |
| **Information Technology** | 0.8 | 0.041 | 0.714 |
| **Materials** | 0.2 | 0.036 | 0.99 |
| **Real estate** | 0.6 | 0.026 | 0.805 |
| **Util** | 0.4 | 0.019 | 0.931 |

series return $\{X_t\}$, $t = 1, \ldots, T$, dot product.

$$\tilde{X}_t = \sum_{k=0}^{\infty} \omega_k X_{t-k} \tag{10}$$

$$\omega_0 = 1, \omega_k = -\omega_{k-1} \frac{d-k+1}{k} \tag{11}$$

where the integer k $\geq$ 0. The fractional differentiated value $\tilde{X}_t$ can be calculated using a fixed-width window by removing the weights when the $|\omega_k|$ lags behind a specified threshold (ξ). Suggestion given above is finding the first $l^*$ to mitigate $|\omega_{l^*}| \geq \xi$ and $|\omega_{l^*+1}| \leq \xi$. From this perspective, we compute the FFD(d) series for d $\in$ [0.1] and find that the lowest d at which the p-value for the Augmented Dickey-Fuller statistics on FFD(d) is less than 5%. The final FFD (d) series is used in this study.

Table 2 shows the calculated results of optimal d to mitigate the stationarity within 5% significance, while maintaining the original data traits as much as possible. The correlation coefficient indicates the similarity from the original dataset after fractional differentiation.

The Fig 1 shows the descriptive statistics after applying optimal FFD for each indexes. We perform Augmented Dickey-Fuller (ADF) test and Jarque-Bera test to ensure the stationarity and normality. All values applying optimal FFD are stationary and normalized, and significant at 5% confidence level.

## Macroeconomic variable and others

This study analyses the relationship between macroeconomic variables (including category-specific Economic Policy Uncertainty (EPU)/ Equity Market Volatility (EMV)) and indices of stock return volatility. The data sources for this study include ten macroeconomic variables, two market sentiment indices, and four category-specific EPU/EMV indices. We collect macroeconomic data from FactSet, a financial data vendor, and the Economic Statistics System (ECOS: https://ecos.bok.or.kr/#/) from the Bank of Korea. Category-specific EPU indices were downloaded from the EPU website (https://www.policyuncertainty.com). Additionally, South Korea is one of the dividend nations after the Korean War armistice agreement; therefore, we

| Indices | Obs. | Mean | Std.Dev | Min | Max | Skenewss | Kurtosis | ADF | Jarque-Bera | Q(5) | Q(10) |
|---|---|---|---|---|---|---|---|---|---|---|---|
| Growth | 3440 | 0.0002787 | 0.0123515 | -0.081141 | 0.0935512 | -0.18926 | 4.718487 | -22.585*** | 3200.088*** | 6.552 | 21.526** |
| Value | 3440 | 0.0002807 | 0.0119802 | -0.107446 | 0.0857459 | -0.364662 | 7.360653 | -22.309*** | 7815.638*** | 15.225*** | 31.339*** |
| LargeMidCap | 3440 | 0.000292 | 0.0120888 | -0.079636 | 0.0874205 | -0.155929 | 5.276573 | -22.456*** | 3990.425*** | 7.486 | 23.997*** |
| SmallCap | 3440 | 0.0002503 | 0.0128312 | -0.122377 | 0.0923963 | -0.85227 | 7.470302 | -22.223*** | 8387.908*** | 19.257*** | 28.080*** |
| Communication Services | 3440 | 0.0002134 | 0.0138587 | -0.071008 | 0.0808696 | 0.206224 | 2.355767 | -57.416*** | 816.325*** | 3.885 | 7.331 |
| Consumer Discretionary | 3440 | 0.0002934 | 0.014314 | -0.102821 | 0.1009383 | -0.089811 | 5.45204 | -58.234*** | 4250.075*** | 3.215 | 10.72 |
| Consumer Staple | 3440 | 0.0001689 | 0.0111143 | -0.067916 | 0.0600829 | -0.234263 | 2.075766 | -34.715*** | 646.203*** | 13.987** | 19.327** |
| Energy | 3440 | 0.0002537 | 0.021266 | -0.148932 | 0.1727345 | 0.216877 | 5.458055 | -30.283*** | 4281.773*** | 7.369 | 14.036 |
| Financials | 3440 | 0.0001532 | 0.0153258 | -0.114216 | 0.1090715 | -0.043667 | 7.093305 | -58.467*** | 7188.425*** | 3.155 | 9.362 |
| Healthcare | 3440 | 0.0004218 | 0.0172503 | -0.134202 | 0.1135307 | -0.343439 | 4.697909 | -22.822*** | 3219.453*** | 12.689** | 22.44** |
| Industrials | 3440 | -1.95E-05 | 0.0150339 | -0.118888 | 0.0970525 | -0.39374 | 6.762087 | -21.727*** | 6620.436*** | 13.335** | 28.434*** |
| Information Technology | 3440 | 0.0004978 | 0.0151484 | -0.06989 | 0.1017374 | 0.079325 | 2.241598 | -23.198*** | 720.611*** | 15.661*** | 25.294*** |
| Materials | 3440 | 0.0002488 | 0.0155454 | -0.140286 | 0.1074897 | -0.29418 | 5.67752 | -22.439*** | 4653.564*** | 9.073 | 23.742*** |
| Real estate | 3440 | 0.0002394 | 0.0203393 | -0.143394 | 0.1397619 | 0.505887 | 9.573139 | -25.642*** | 13239.462*** | 7.46 | 20.965** |
| Util | 3440 | 2.767E-06 | 0.0162465 | -0.082736 | 0.1025579 | 0.308854 | 2.756653 | -31.282*** | 1139.314*** | 18.244*** | 26.556*** |

**Fig 1. Descriptive statistics on South Korea benchmark indices.**

employ the Geopolitical Risk index (GPR) of geopolitical risk from North Korea [17]. Every dataset included the sample period from January 2009 to September 2022 and was collected monthly after eliminating seasonal factors.

Fama [38] argued that expected growth in nominal money supply is positively related to stock return volatility. Rahmi et al. [39] and Amihud [40] studied the impact of money supply and liquidity indicators on stock volatility such as Negotiable certificate of deposit, M1/M2, Investment Balance. Therefore, we use the month-on-month growth rate of money supply (M1); a broader definition of money includes everything in M1 but also adds other types of deposits (M2). 91 days CD rate is used as the interest rate variable.

Ilmanen [41], Engle et al. [19], Aslanidis and Christiansen (2016), and Flannery [42] argued that general economic factors are related to stock return volatility. We used the producer price index (PPI), unemployment rate, retail sales index, housing and construction building permits by residential usage, house sales price index, and exchange rate (KRW/USD) as overall economic variables.

The Business Survey Index (BSI) reflects entrepreneurs' views and expectations of the economy, and significantly impacts production, sales, and investment. The Economic Sentiment Index (ESI) is a composite index of the BSI and the consumer trend index that selects seven highly responsive economic items. Indices above 100 indicate economic improvement, whereas those below 100 indicate predicted economic deterioration.

The relationship between monetary policy and stock volatility has been widely debated [43]. Borio and White [44] argued that monetary policies can significantly affect asset price volatility. Volatility forecasting considers the monthly realized volatility of each index [45]. All the monthly variables used in this study are transformed into natural log differences. If the original data are not a rate, we multiply it by 100. The Fig 2 shows the descriptive statistics for macroeconomic variables.

| Variables | Obs. | Mean | Std.Dev | Min | Max | Skewness | Kurtosis | ADF | Jarque-Bera | Q(5) | Q(10) |
|---|---|---|---|---|---|---|---|---|---|---|---|
| *PPI* | 165 | 0.0014 | 0.0052 | -0.0123 | 0.0161 | 0.1438 | 0.4474 | -6.673*** | 1.646 | 96.535*** | 134.435*** |
| *UNEMP* | 165 | -0.0012 | 0.0711 | -0.2877 | 0.2666 | -0.3511 | 3.4932 | -6.259*** | 80.58*** | 18.808*** | 46.589*** |
| *M1* | 165 | 0.0086 | 0.0107 | -0.0244 | 0.044 | 0.0431 | 0.5598 | -2.829* | 1.816 | 40.306*** | 56.873*** |
| *M2* | 165 | 0.0059 | 0.0045 | -0.0125 | 0.0186 | -0.006 | 1.1129 | -3.249** | 7.485** | 14.370** | 26.411*** |
| *RET_SALE* | 165 | 0.0027 | 0.02 | -0.0784 | 0.0502 | -0.6045 | 2.1716 | -10.639*** | 39.333*** | 32.049*** | 38.466*** |
| *BSI* | 165 | -0.0006 | 0.0141 | -0.0388 | 0.047 | -0.0438 | 0.3775 | -5.909*** | 0.801 | 15.006** | 32.664*** |
| *91DAYS CD RATE* | 165 | -0.0027 | 0.0655 | -0.3739 | 0.2412 | -1.0617 | 8.2603 | -5.74*** | 467.793*** | 90.624*** | 101.731*** |
| *NHC* | 165 | -0.0003 | 0.1101 | -0.4291 | 0.3106 | -0.1898 | 1.1845 | -5.957*** | 9.485*** | 25.89*** | 34.856*** |
| *H_SALE* | 165 | 0.0015 | 0.0039 | -0.0074 | 0.0151 | 0.6644 | 1.1445 | -3.397** | 19.849 | 359.416*** | 422.06*** |
| *CPI* | 165 | 0.0017 | 0.0027 | -0.0047 | 0.0088 | 0.1923 | 0.1191 | -3.373** | 1.042 | 18.167*** | 30.209*** |
| *ESI* | 165 | 0.0014 | 0.0153 | -0.045 | 0.0571 | 1.06 | 3.5429 | -4.491*** | 109.829*** | 403.408*** | 415.14*** |
| *K_EPU* | 165 | 0.0004 | 0.1399 | -0.3732 | 0.4566 | 0.1202 | 0.5765 | -10.92*** | 2.271 | 19.607*** | 24.14*** |
| *US_EPU* | 165 | 0.0015 | 0.2844 | -0.9163 | 0.891 | 0.0442 | 0.6976 | -9.833*** | -2.873 | 20.515*** | 24.413*** |
| *F_EMV* | 165 | -0.0061 | 0.4723 | -1.4231 | 2.098 | 0.6838 | 2.5643 | -8.712*** | 53.931 | 27.659*** | 32.384*** |
| *OVERALL_EMV* | 165 | -0.0034 | 0.278 | -0.8236 | 0.9525 | 0.1608 | 1.0596 | -6.975*** | 7.459* | 16.481*** | 22.533** |
| *GPRI* | 165 | 0.0025 | 0.1826 | -0.5085 | 0.7244 | 0.2877 | 0.6969 | -8.211*** | 5.049* | 22.096*** | 23.316*** |
| *EX_RATE* | 165 | 0.0008 | 0.0289 | -0.1033 | 0.1062 | 0.3194 | 2.694 | -13.219*** | 48.408 | 3.033 | 11.004 |

**Fig 2. Descriptive statistics on macroeconomic variables.**

## Empirical results

### In-sample analysis

We varied the tuning parameter λ from 0 to 15 in increments of 0.2. We take the optimal λ that makes the minimum GIC. However, some indices do not converge until 15; therefore, we extend the number of data points from 15 to 30, which increases the number of data points by more than two times. During the estimation, we use the restricted beta-weighting scheme, and the lag parameter K in Eq (6) is determined to be 24 for the monthly data [46, 47]. Using the adaptive lasso regression, we find the most significant macroeconomic variables that affect stock volatility. The more the penalized parameter λ increases, the more the regression bias increases, and unimportant variables are removed. We conduct an in-sample analysis using the following steps:

1. Estimate the parameter $\widehat{\theta}_j$ and $\Phi_2$ using the GARCH-MIDAS model with all 18 variables and calculate the adaptive weights $\widehat{\omega}_j = 1/|\widehat{\theta}_j|^\eta$.

2. Conduct the GARCH-MIDAS model with feature selection by fixing the condition using the estimated parameters in step (1). The PLLF in Eq (7) is calculated with the hyper-parameter λ on a range of [0, 15]. If the number of converged variables remains over 50% then the λ range is extended from 15 to 30

3. Calculate the GIC by Eq (8) and determine the optimal λ that minimizes the GIC. Finally, determine the most influential variables.

4. As a subsequent selection step, the GARCH-MIDAS model is estimated with the chosen variables to determine their significance. Since the estimated θ represents the most

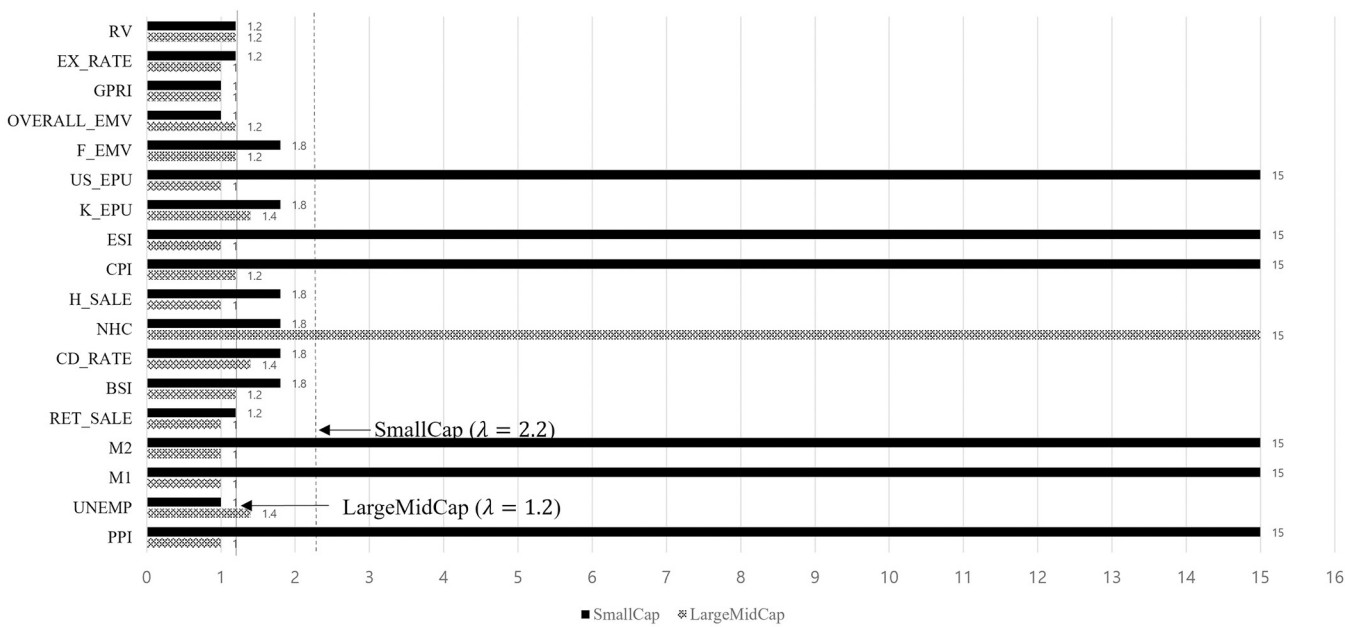

**Fig 3. The size effects for each variable as the coefficient decrease to zero with increasing lambda.**

considerable value of long-term stock volatility, this process serves to verify the significance of the selected variables.

**Size effects analysis.** The LargeMidCap index comprises 163 equities, and the IT and Industrial sectors account for over 50% of all aggregate weights for the index. The influence subset variables are almost the same as the union subset of the IT and Industrial sectors.

As shown in Fig 3, the optimal λ for LargeMidCap and SmallCap indices is 1.2 and 2.2 respectively. For the LargeMidCap index, the unemployment rate, BSI, CD rate, new housing construction, Consumer Price Index (CPI), Korea EPU, financial crisis EMV, overall EMV and realized volatility are the dominant factors. The SmallCap index is influenced by PPI, M1, M2, CPI, ESI, and US EPU. After determining the influencing variables, we perform a post-selection estimation. We are interested in the most significant variables to the stock volatility, so the estimated $\theta$ is the prominent.

The post-selection estimation results for LargeMidCap and SmallCap are presented in Table 3. All the estimated parameters, except the overall EMV, are significant at the 1%

**Table 3. Size effects analysis estimation after feature selection.**

| LargeMidCap | $\theta_{UNEMP}$ | $\theta_{BSI}$ | $\theta_{CD\_RATE}$ | $\theta_{NHC}$ | $\theta_{CPI}$ |
|---|---|---|---|---|---|
| | 37.303*** (12.158) | 0.729*** (0.278) | -21.984*** (3.301) | -22.118*** (6.132) | 3.36*** (0.944) |
| | $\theta_{K\_EPU}$ | $\theta_{F\_EMV}$ | $\theta_{OV\_EMV}$ | $\theta_{RV}$ | |
| | 11.63*** (3.346) | -16.6*** (4.202) | 0.291 (0.224) | 14.522*** (2.724) | |
| SmallCap | $\theta_{PPI}$ | $\theta_{M1}$ | $\theta_{M2}$ | $\theta_{CPI}$ | $\theta_{ESI}$ |
| | 5.044*** (1.148) | 1.646*** (0.194) | 2.197*** (0.261) | -7.383*** (1.152) | 0.169*** (0.052) |
| | $\theta_{US\_EPU}$ | | | | |
| | 21.293*** (4.886) | | | | |

*** means significant at a 1% confidence level. The numbers in parentheses are robust standard errors.

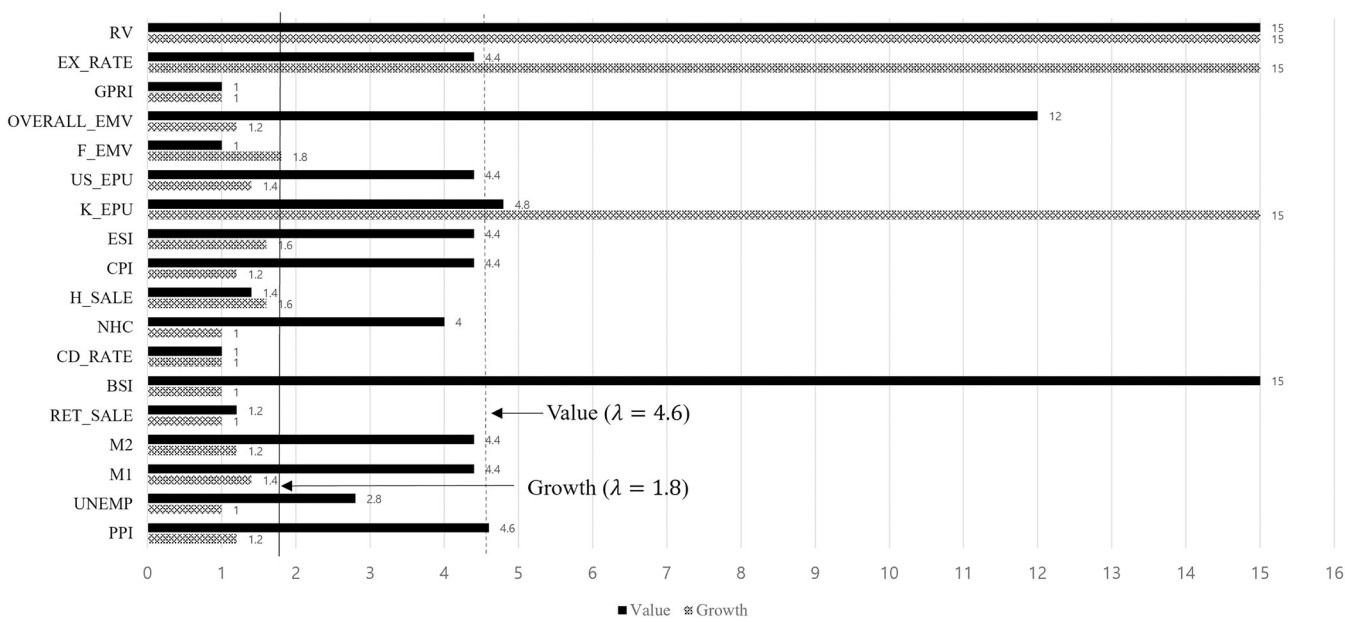

**Fig 4. Style effects on LargeMidCap and SmallCap.**

confidence level. For LargeMidCap, the estimated unemployment rate is 37.303. A higher unemployment rate leads to higher future large midcap stock volatility. Historically, bad news or worse unemployment rates are usually good for stocks [48]. Business confidence has idiosyncratic features; therefore, stock market responses may change in each country [49]. Normally, this has a positive effect on stock volatility because if owners have strong confidence in the business environment, they are willing to invest more. This, in turn, increases stock volatility.

The estimated BSI is 0.729, which means that increasing BSI leads to higher long-term LargeMidCap volatility. Clearly, a company's procurement costs increase as the interest rate increases. Additionally, it degrades investment attraction to stocks. Hence, the estimated CD rates negatively affect stock volatility. Usually, major market participants are interested in large- or medium-sized companies, which have negative effects. Similar to a previous study on US stocks [50], the estimated parameter for new housing construction is -22.118, meaning that new housing construction leads to lower long-term LargeMidCap volatility. The CPI, Korea EPU, overall EMV and realized volatility is associated with long-term LargeMidCap volatility. Interestingly, CPI has a different effect on LargeMidCap and SmallCap. The higher the inflation rate, the larger the LargeMidCap size of volatility. Another interesting point is that SmallCap size is influenced by US EPU rather than Korea EPU, similar to LargeMidCap. With the development of mass media and communication technologies, SmallCap has become the main investment target for individual investors. Usually, individual investors, especially in the US stock market, are easily affected by uncertainty. Hence, the long-term volatility of SmallCap is positively influenced by US EPU. Overall, the LargeMidCap stock is affected by the overall macroeconomic environmental status, political uncertainty, and even a wide range of financial crises. In contrast, the SmallCap stock is influenced by money supply and overall market sentiment.

**Style effects analysis.** Growth stocks are typically considered to outperform the overall market because of their potential, whereas value stocks are traded based on their worth. Fig 4 shows style effects for each variable as the coefficient decreases to zero with increasing lambda.

**Table 4. Style effects analysis estimation after feature selection.**

| Growth | $\theta_{K\_EPU}$ | $\theta_{F\_EMV}$ | $\theta_{EX\_RATE}$ | $\theta_{RV}$ | |
|---|---|---|---|---|---|
| | 10.978*** (0.541) | 1.421*** 0.116) | 11.739*** (0.253) | 3.681*** (0.661) | |
| Value | $\theta_{PPI}$ | $\theta_{BSI}$ | $\theta_{K\_EPU}$ | $\theta_{OVERALL\_EMV}$ | $\theta_{RV}$ |
| | 1.847*** (0.023) | 0.269*** (0.009) | 20.626*** (0.197) | 16.743*** (0.119) | 1.93*** (0.017) |

*** means significant at a 1% confidence level. The numbers in parentheses are robust standard errors.

Value stocks are usually large well-established companies. Hence, the selected variables are included in the selected variable list of LargeMidCap. However, growth stocks can be small-, mid-, and large-cap, so they are influenced by the overall variables; however, the Korea EPU and exchange rate are the most dominant factors. The Realized Volatility (RV) is a common factor that affects stock volatility. Table 4 presents the post-selection estimation results for the style effect.

A limitation is that the growth index consists of the mixture capital size, which we do not consider in this study because of data availability. However, we find that growth stocks are sensitive to shocks in Korea EPU or the financial crisis EMV [51]. The exchange rate is a key factor in increasing stock volatility in the South Korean market. Foreign investors tend to sell when the exchange change rate increases or when it is expected to increase to minimize the loss caused by the currency exchange. This increases stock volatility. Normally, growth stocks are overvalued compared to their performance, so they sell first in the status of currency exchange risk. The selected variables for value stocks are similar to those for LargeMidCap. The high inflation rate usually leads to high stock volatility [19], in this manner, a high PPI follows the soaring stock volatility on value stocks. The EMV tracker is constructed based on economic or financial uncertainty or risk. Uncertainty and risk have greater volatility effects.

**Sector effects analysis.** Macroeconomic variables affect 11 sectors differently. However, an interesting finding is that the housing price-sales index and realized volatility are commonly effective variables in most sectors. Fig 5 shows the sector effects on the consumer staples, consumer discretionary, and communication services sectors for each variable as the coefficient decreases to zero with increasing lambda.

The communication services sector is affected by M1, housing price-sales index, CPI, financial crisis, EMV, and realized volatility. BSI, US EPU, overall EMV, and realized volatility are the selected variables for the consumer discretionary sector. However, the consumer staple sector is influenced by M2, new housing construction, the housing price-sales index, and Korea EPU.

Table 5 shows the post-selection estimation results for the communications services, consumer staples, and consumer discretionary sectors. In Korea, the communication services sector is treated as a defensive stock, so volatility decreases compared to other cyclical stocks when the money supply increases. This reveals that the estimated $\theta_{M1}$ is -0.379, which means that the decreasing volatility in communication services sector stocks follow an increased money supply. Housing prices normally increase when the economy is well-shaped. If the economy is good, the volatility of a defensive stock will decrease as the proportion of investment in economy-sensitive stocks increases, rather than in economy-defensive stocks.

Thus, there is a negative relationship between the housing price-sales index and the CPI, which are indicators of the economy. The major financial crises in the research period were the US government debt-ceiling crisis (2011.08) and the Chinese stock market crashes from June 2014 to June 2015 [11]. Financial crises increase short-term volatility but tend to reduce

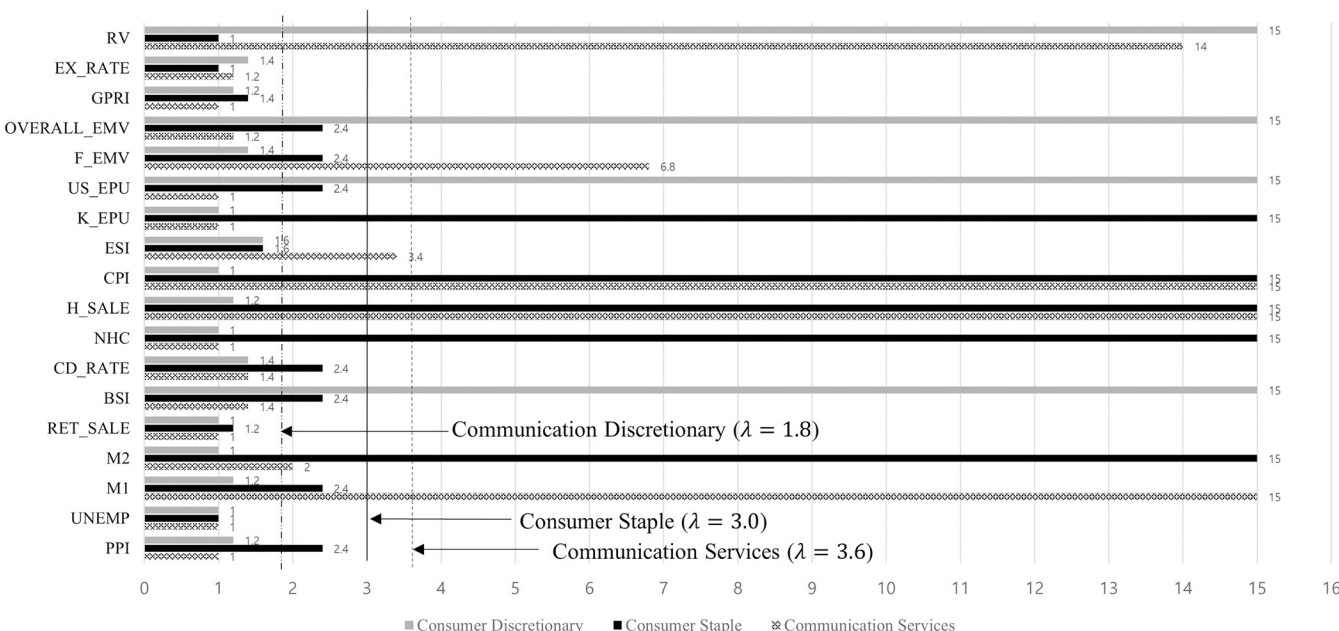

**Fig 5. Sector effects on communication services, consumer discretionary, and consumer staple sectors.**

long-term volatility by triggering a preference for risk-free assets. Similarly, the estimated $\theta_{F\_EMV}$ for the communications services sector is negative.

Consumer staples are daily necessities, whereas consumer discretionary goods are nonessential items bought when the economy is strong. An increase in M2, which indicates a growing economy, can increase volatility in the consumer staples industry. Conversely, a rise in the housing sales price index, which reflects a robust housing market, can reduce volatility in this sector. However, new housing construction influenced by government policies can increase volatility. The consumer discretionary sector is affected by the BSI, US EPU Index, overall EMV, and realized volatility. High BSI and US EPU can lead to increased volatility in this sector.

Fig 6 shows the effects on the IT, financial, and energy sectors for each variable as the coefficient decreases to zero with increasing lambda. The IT sector is influenced by PPI, M1, M2, and new housing construction. The financial sector is affected by the housing price-sales index, CPI, ESI, and realized volatility. Additionally, M2, the BSI, CD rate, and US EPU affect the energy sector.

**Table 5. Sector effects analysis estimation after feature selection—Communication services, consumer discretionary and consumer staple sectors.**

| **Communication Services** | $\theta_{M1}$ | $\theta_{H\_SALES}$ | $\theta_{CPI}$ | $\theta_{F\_EMV}$ | $\theta_{RV}$ |
|---|---|---|---|---|---|
| | -0.379*** (0.653) | -0.908*** (0.347) | -5.262*** (2.521) | -24.332*** (8.901) | 9.199*** (2.033) |
| **Consumer Discretionary** | $\theta_{BSI}$ | $\theta_{US\_EPU}$ | $\theta_{OV\_EMV}$ | $\theta_{RV}$ | |
| | 0.31* (0.165) | -1.907*** (0.046) | 11.878*** (3.865) | 7.805*** (1.226) | |
| **Consumer Staple** | $\theta_{M2}$ | $\theta_{NHC}$ | $\theta_{H\_SALE}$ | $\theta_{OV\_EMV}$ | $\theta_{RV}$ |
| | 1.985*** (0.116) | 4.946*** (0.704) | -1.146*** (0.14) | -1.45*** (0.4) | -1.757*** (0.257) |

***,**,* means significant at 1%, 5% and 10% confidence level, respectively. The numbers in parentheses are robust standard errors.

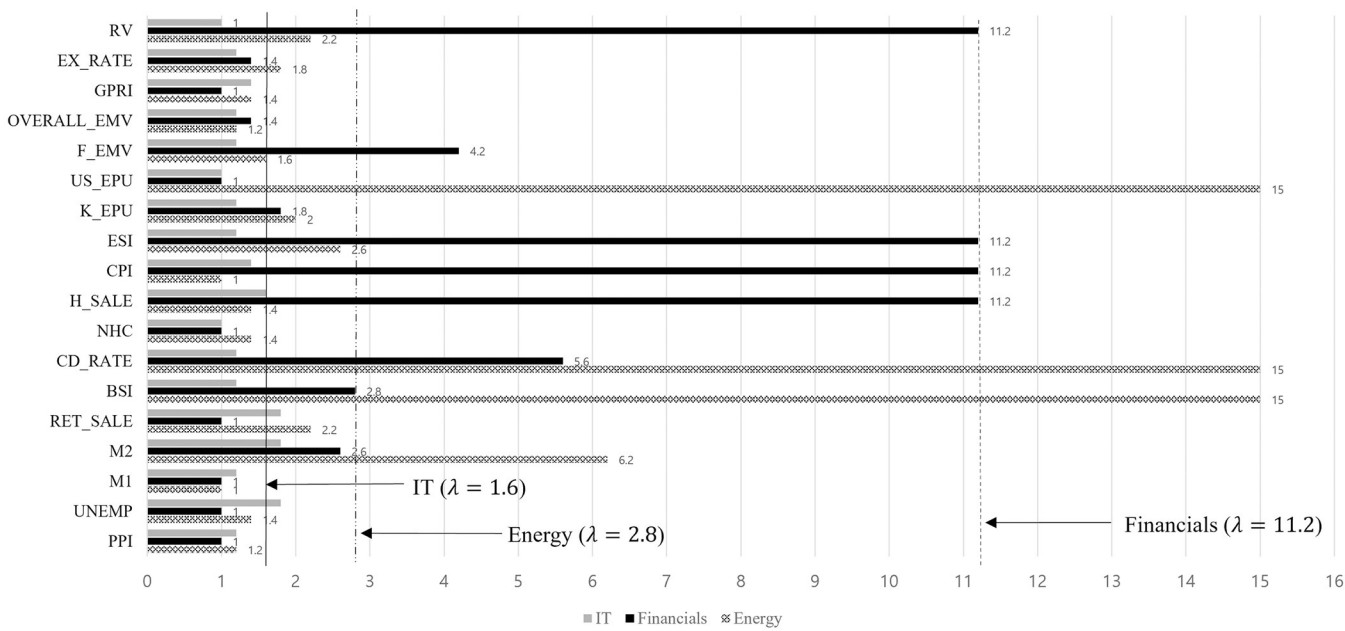

**Fig 6. Sector effects on energy, financials, and IT.**

Table 6 presents the post-selection estimation results for the IT, financial, and energy sectors. Money supply indicators affect the volatility of IT sector stocks. This finding is similar to that of previous studies [39]. An increase in the PPI may indicate inflation, leading to higher interest rates and borrowing costs, which could increase IT sector volatility. However, an increase in new housing construction, often due to eased regulations during economic downturns, can decrease volatility in the IT sector.

In the financial sector, an increase in housing sales prices and the CPI can increase volatility because of the sector's close ties with the real estate industry. Interestingly, only the consumer staples and financial sectors are negatively affected by realized volatility. Improvements in the ESI can reduce financial sector volatility by encouraging investments in promising industries. The estimated parameters on 91 Days CD rate and US EPU have a positive effect on the volatility of energy sector stocks. In contrast, M2 and BSI negatively impact energy sector stock. This uncertainty increases energy market volatility, similar to the findings of a previous study [52]. Particularly, regulatory uncertainty in the energy sector in the US causes fluctuations in global energy prices, resulting in high volatility. Growth in M2 and a positive BSI indicate a strong economic outlook and money flows into more promising future industries. Fig 7 shows the

**Table 6. Sector effects analysis estimation after feature selection—IT, financials and energy.**

| IT | $\theta_{PPI}$ | $\theta_{M1}$ | $\theta_{M2}$ | $\theta_{NHC}$ |
|---|---|---|---|---|
| | -1.406 *** (0.105) | 0.512*** (0.158) | 3.299*** (0.604) | -35.671*** (2.747) |
| **Financials** | $\theta_{H\_SALE}$ | $\theta_{CPI}$ | $\theta_{ESI}$ | $\theta_{RV}$ |
| | 3.764*** (0.165) | 8.941*** (0.188) | -1.376*** (0.071) | -10.565*** (0.229) |
| **Energy** | $\theta_{M2}$ | $\theta_{BSI}$ | $\theta_{CD\_RATE}$ | $\theta_{US\_EPU}$ |
| | -2.792*** (0.033) | -3.235*** (0.024) | 8.162*** (0.561) | 8.998*** (0.18) |

***,**,* means significant at 1%, 5% and 10% confidence level, respectively. The numbers in parentheses are robust standard errors.

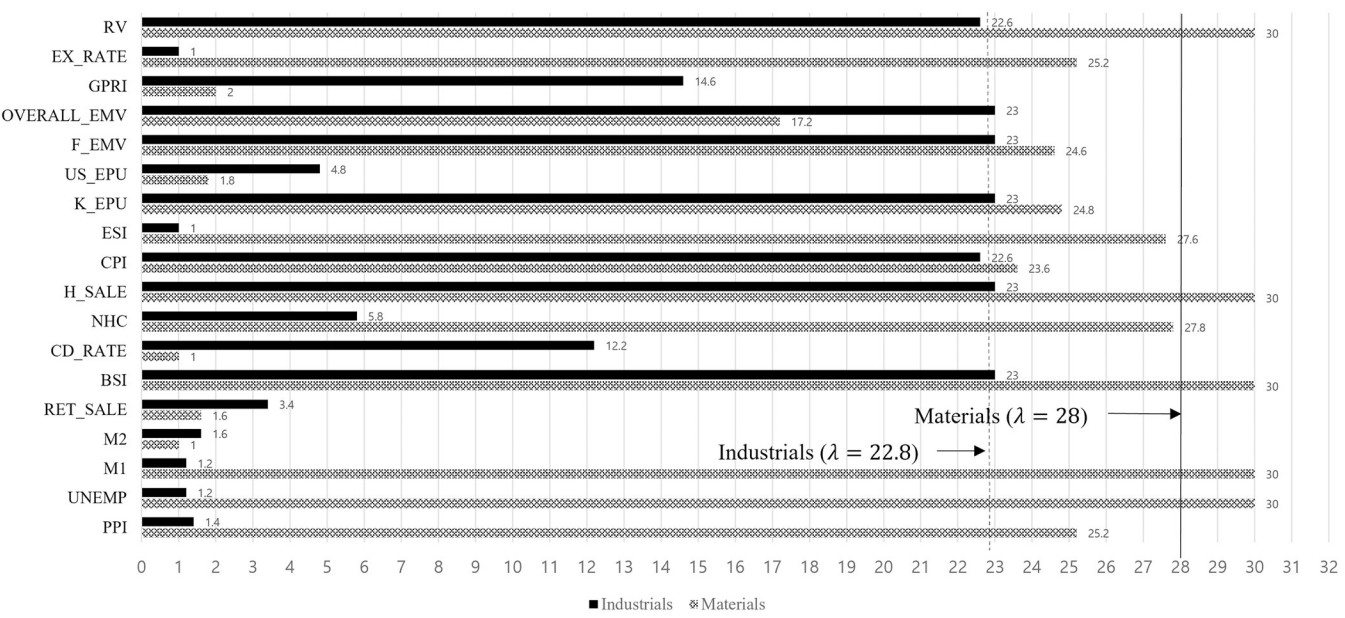

**Fig 7. Sector effects on industrials and materials.**

selected variables in the optimal λ to make minimum GIC as the coefficient decrease to zero with increasing lambda on materials and industrial sectors.

These two sectors are insufficient to converge in the range of [0, 15] for $\lambda_{max}$ so we extend the maximum range to 30. The selected variables for the materials are the unemployment rate, M1, BSI, housing price-sales index, and realized volatility. With the increasing of λ in industrials, BSI, housing price-sales index, KR EPU, financial crisis EMV, overall EMV and realized volatility.

As Table 7 shows, the industrial sector is positively influenced by uncertainty indicators, such as Korea EPU (8.157), the financial crisis's EMV (22.981), and the overall EMV (2.103). These variables lead to greater volatility in the industrial sector stock. In contrast, the BSI and the housing price-sales index lead to reduced volatility. All the selected variables are significant at the 1% confidence level. The estimated parameters of the unemployment rate (48.534), BSI (1.671), housing price-sales index (2.356) and realized volatility (8.644) positively impact the volatility of materials sector stocks. However, M1 (-0.574) leads to downward volatility. All the selected variables are also significant at the 1% confidence level.

**Table 7. Sector effects analysis estimation after feature selection–industrials and materials.**

| Industrials | $\theta_{BSI}$ | $\theta_{H\_SALE}$ | $\theta_{K\_EPU}$ | $\theta_{F\_EMV}$ | $\theta_{OV\_EMV}$ |
|---|---|---|---|---|---|
| | -3.697*** (0.58) | -4.693*** (0.477) | 8.157*** (0.779) | 22.981*** (5.112) | 2.103*** (0.755) |
| | $\theta_{RV}$ | | | | |
| | 20.247*** (0.711) | | | | |
| Materials | $\theta_{UNEMP}$ | $\theta_{M1}$ | $\theta_{BSI}$ | $\theta_{H\_SALE}$ | $\theta_{RV}$ |
| | 48.534*** (14.677) | -0.574*** (0.27) | 1.671*** (0.84) | 2.356*** (0.534) | 8.644*** (2.318) |

*** means significant at a 1% confidence level. The numbers in parentheses are robust standard errors.

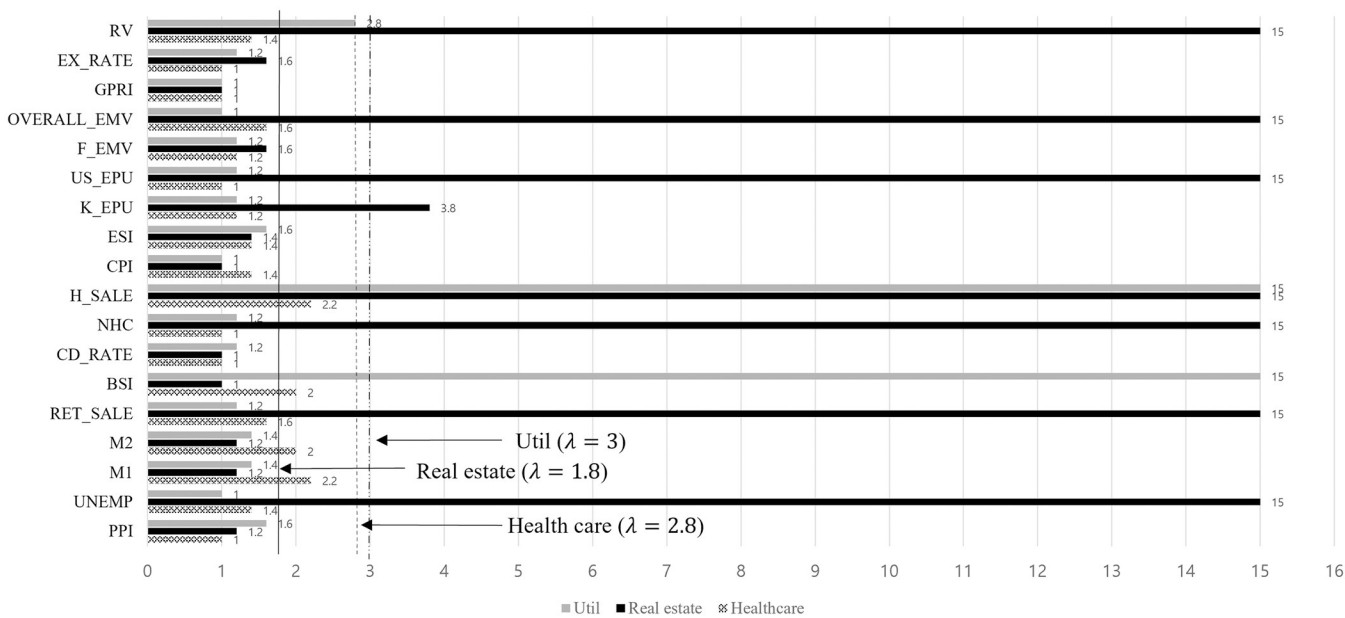

**Fig 8. Sector effects on healthcare, real estate and util.**

Fig 8 shows the selected variables in the optimal λ for util, real estate and healthcare sectors. Surprisingly, there are no remaining variables in the healthcare sector at the minimum GIC.

The healthcare industry in South Korea is in its infancy, so most stocks are mainly influenced by individual events. Therefore, we cannot find a significant long-term volatility impact factor. However, we take the variables which converge to the nearest optimal λ. The selected variables for util sector are the BSI, housing price-sales index, and realized volatility. The real estate sector is influenced by the unemployment rate, retail sales index, housing price-sales index, new housing construction, Korea EPU, financial crisis EMV, overall EMV, and realized volatility. The nearest variables to the optimal λ for the healthcare sector are M1 and housing price sales price. Table 8 presents the post-selection estimation results on util, real estate and healthcare sectors.

The util sector is negatively associated with BSI and the housing price-sales index. The estimated $\theta_{BSI}$ and $\theta_{H\_SALE}$ are -0.995 and -4.015, respectively and they are significant at the 1%

**Table 8. Sector effects analysis estimation after feature selection–util, real estate and healthcare.**

| Util | $\theta_{BSI}$ | $\theta_{H\_SALE}$ | $\theta_{RV}$ | | |
|---|---|---|---|---|---|
| | -0.995*** (0.389) | -4.015*** (0.994) | 0.904*** (0.41) | | |
| **Real estate** | $\theta_{UNEMP}$ | $\theta_{RET\_SALE}$ | $\theta_{H\_SALE}$ | $\theta_{NHC}$ | $\theta_{K\_EPU}$ |
| | 47.827*** (4.126) | -36.31*** (4.126) | 9.368*** (0.541) | -0.609*** (0.284) | -3.595*** (0.173) |
| | $\theta_{F\_EMV}$ | $\theta_{OV\_EMV}$ | $\theta_{RV}$ | | |
| | 2.703*** (0.151) | -14.478*** (1.325) | 7.136*** (0.135) | | |
| **Healthcare** | $\theta_{M1}$ | $\theta_{H\_SALE}$ | | | |
| | 0.785 (-) | 1.0 (-) | | | |

*** means significant at a 1% confidence level. The numbers in parentheses are robust standard errors.

level. A positive BSI and housing price-sales index indicate a strong economy. In South Korea, the util sector is the basic industry. Therefore, it is less preferable than other promising future industries, leading to downward volatility. The estimated $\theta_{UNEMP}$ on the real estate sector is 47.827, which is significant at the 1% level. The increasing unemployment rate increases the volatility of real estate sector stocks. The retail sales index, new housing construction, KR EPU, and overall EMV decrease volatility, but the housing price-sales index and financial crisis EMV decrease the volatility of real estate sector stocks. Finally, we perform the post-selection estimation for the healthcare sector with the two variables which take the nearest optimal λ. However, the differences are not statistically significant. Realized volatility positively correlates with overall market volatility. We reconfirm previous findings that overall financial volatility has a countercyclical pattern [19, 47].

## Out-of-sample forecasting exercise

We evaluate the predicted volatility to compare with the true volatility and out-of-sample volatility. The out-of-sample forecasting period is from October 2022 to December 2022 (three months). There is a limitation in gathering some macro variables. Therefore, the forecasting period is somewhat insufficient to confirm the effectiveness of long-term volatility with the selected macro variables. We calculate the root mean squared forecast error (RMSE) using a rolling window method to forecast 1-month, 2-months, and 3-months ahead volatility.

$$\text{RMSE} = \sqrt{\frac{1}{N}\sum_{n=1}^{N}\left(\sigma_{n+k}^2 - \widehat{\sigma}_{n+k}^2\right)^2} \tag{12}$$

The models compared are basic GARCH models as a benchmark model. We calculate $\text{RMSE}_{Benchmark}$/RMSE to measure performance, and a ratio greater than 1 signifies that our baseline model performs more effectively. Table 9 shows that the model selected for this study performs significantly better than most of the compared models. However, in a few cases, the chosen model is worse than the basic GARCH model. The SmallCap, IT, and healthcare sectors show lower performance than the basic model for SmallCap stocks, and an imbalance in information availability often results in greater volatility when new information or news enters the market. Furthermore, investor panic selling and mass purchasing can lead to drastic volatility. Notably, the abundance of liquidity and sudden changes in information and social systems following the COVID-19 pandemic have increased short-term volatility. Particularly, rapid digitalization after COVID-19 and the development of the Fourth Industrial Revolution have led to increased short-term volatility in the IT and healthcare sectors. This likely diminishes the predictive power of macroeconomic variables. In future research, an analysis of the negative or positive impacts of major news on stock prices for individual stocks and sectors could potentially enhance predictive accuracy.

## Discussion and conclusion

This study investigates the relationship between macroeconomic variables and stock volatility across capital sizes, styles, and sectors in South Korea through an analysis of S&P South Korea BMI LargeMidCap, SmallCap, Growth, and Value, and 11 Sectors Index values from January 2009 to September 2022. This study uses the GARCH-MIDAS model with feature selection to explore how macroeconomic variables affect the South Korean stock market by employing an optimal fractionally differentiated price series for maximum memory preservation. The findings show that large-and midcap stocks are influenced by overall economic conditions, whereas SmallCap stocks are significantly affected by money supply (M1 and M2). The CPI and RV are common influencing factors. This suggests investors might consider adjusting

**Table 9. Out-of-sample prediction results.**

| Variable Tenor | RMSE | | |
|---|---|---|---|
| | **1-month** | **2-months** | **3-months** |
| *Panel A: Style Effects Analysis* | | | |
| *Panel A-1: Growth* | | | |
| **Basic GARCH Model** | 7.2094* | 8.4139* | 6.2912* |
| **Selected Variables** | 6.8579 | 7.9447 | 7.51 |
| **RMSE$_{Benchmark}$/RMSE** | **1.0513*** | **1.0591*** | **1.0257*** |
| *Panel A-2: Value* | | | |
| **Basic GARCH model** | 20.5685* | 30.8452* | 28.3704* |
| **Selected Variables** | 12.7749 | 20.6247 | 16.5509 |
| **RMSE$_{Benchmark}$/RMSE** | **1.6101*** | **1.4955*** | **1.71418*** |
| *Panel B: Size Effects Analysis* | | | |
| *Panel B-1: LargeMidCap* | | | |
| **Basic GARCH model** | 20.2959* | 25.7656* | 22.3019* |
| **Selected Variables** | 1.4531 | 1.2224 | 1.2215 |
| **RMSE$_{Benchmark}$/RMSE** | **13.9673*** | **21.0779*** | **18.2578*** |
| *Panel B-2: SmallCap* | | | |
| **Basic GARCH model** | 11.7001* | 12.7782 | 11.3417 |
| **Selected Variables** | 11.6422 | 17.4834 | 11.6246 |
| **RMSE$_{Benchmark}$/RMSE** | **1.005*** | 0.7309 | 0.9757 |
| *Panel C: Sector Effects Analysis* | | | |
| *Panel C-1: Consumer Staples Sector* | | | |
| **Basic GARCH model** | 14.5358* | 11.7261* | 10.8557* |
| **Selected Variables** | 11.2902 | 7.0586 | 5.9618 |
| **RMSE$_{Benchmark}$/RMSE** | **1.2875*** | **1.6616*** | **1.8201*** |
| *Panel C-2: Consumer Discretionary Sector* | | | |
| **Basic GARCH Model** | 32.7995* | 25.9355* | 34.2014* |
| **Selected Variables** | 17.8107 | 22.4454 | 22.1131 |
| **RMSE$_{Benchmark}$/RMSE** | **1.8416*** | **1.1555*** | **1.5467*** |
| *Panel C-3: Communication Services Sector* | | | |
| **Basic GARCH Model** | 2.2005* | 1.9985* | 1.8278* |
| **Selected Variables** | 2.1254 | 1.8241 | 1.6393 |
| **RMSE$_{Benchmark}$/RMSE** | **1.0353*** | **1.0956*** | **1.115*** |
| *Panel C-4: Information Technology Sector* | | | |
| **Basic GARCH Model** | 11.5227* | 11.3487 | 10.9202 |
| **Selected Variables** | 11.9017 | 13.0773 | 12.389 |
| **RMSE$_{Benchmark}$/RMSE** | 0.9682 | 0.8678 | 0.8814 |
| *Panel C-5: Financial Sector* | | | |
| **Basic GARCH Model** | 8.3251* | 15.6668* | 20.0536* |
| **Selected Variables** | 6.0275 | 12.3608 | 17.8507 |
| **RMSE$_{Benchmark}$/RMSE** | **1.3812*** | **1.2675*** | **1.1234*** |
| *Panel C-6: Energy Sector* | | | |
| **Basic GARCH Model** | 40.48* | 60.5348* | 53.6694* |
| **Selected Variables** | 31.2504 | 57.0416 | 50.4467 |
| **RMSE$_{Benchmark}$/RMSE** | **1.2953*** | **1.0612*** | **1.0639*** |
| *Panel C-7: Industrial Sector* | | | |
| **Basic GARCH Model** | 22.8897* | 19.2483* | 21.0963* |
| **Selected Variables** | 8.6834 | 6.084 | 6.002 |

*(Continued)*

**Table 9.** (Continued)

| Variable Tenor | RMSE | | |
|---|---|---|---|
| | **1-month** | **2-months** | **3-months** |
| RMSE$_{Benchmark}$/RMSE | **2.636*** | **3.1638*** | **3.5149*** |
| *Panel C-8: Material Sector* | | | |
| **Basic GARCH Model** | 45.7315* | 75.7976* | 71.1789* |
| **Selected Variables** | 42.1245 | 63.5281 | 58.5163 |
| RMSE$_{Benchmark}$/RMSE | **1.0856*** | **1.1931*** | **1.2163*** |
| *Panel C-9: Healthcare Sector* | | | |
| **Basic GARCH Model** | 7.2034 | 6.501* | 6.0737 |
| **Selected Variables** | 9.0904 | 6.269 | 7.0202 |
| RMSE$_{Benchmark}$/RMSE | 0.7924 | **1.037*** | 0.8651 |
| *Panel C-10: Real Estate Sector* | | | |
| **Basic GARCH Model** | 5.4129* | 4.5631* | 4.2971* |
| **Selected Variables** | 3.5677 | 3.8279 | 4.1906 |
| RMSE$_{Benchmark}$/RMSE | **1.5172*** | **1.1921*** | **0.8751** |
| *Panel C-11: Util Sector* | | | |
| **Basic GARCH Model** | 8.0941* | 6.9248* | 6.5831* |
| **Selected Variables** | 5.1175 | 3.6573 | 3.1125 |
| RMSE$_{Benchmark}$/RMSE | **1.5817*** | **1.8934*** | **2.1151*** |

A * sign indicates that it was worse than that of the model selected in this study.

their investment strategies across stock sizes during different economic cycles. For instance, a focus on SmallCap stocks could be beneficial during periods of monetary expansion. The significant impact of the housing price-sales index on sector volatility, particularly in real estate-heavy sectors, suggests that sector-specific strategies are critical. Investors should monitor housing market trends for potential shifts in sector performance.

Nextly, the major variable selected for growth stocks is uncertainty. Moreover, financial crisis, EMV, and exchange rate are explanatory variables. This explains why major global events boost volatility in the South Korean stock market. In contrast, value stocks are influenced by overall business conditions. The Korea EPU is a common factor for both value and growth stocks. This suggests that the sensitivity of growth stocks to uncertainty and value stocks to business conditions provides a strategic insight for investors considering style-based investing. The diversification of portfolios to include a mix of growth and value stocks can potentially be a hedge against market volatility due to macroeconomic changes.

Finally, in Korea, a high proportion of real estate assets amplify the volatility impact of the housing price-sales index across most sectors. The housing prices significantly influence stock market volatility in Korea is underpinned by the wealth effect theory and the credit channel theory. The wealth effect posits that changes in housing prices impact consumer wealth and spending, which in turn affects stock markets [53]. In addition, the credit channel theory suggests that housing market values affect borrowing capacity, which in turn affects investment behaviors in the context of stock volatility [54]. Empirical studies, such as Kim work on the Korean market [55]. Our results are consistent with the above theoretical framework and Kim's empirical results on the Korean market, confirming the link between theory and practice. Realized volatility is a common leading indicator and the EMV tracker is useful for predicting stock volatility. Sectors such as communication services, consumer discretionary, and industrials are influenced by the overall EMV or financial crisis EMV. The money supply

indicators M1 and M2 affect sectors, such as IT and healthcare. ESI or BSI impacts sectors like consumer discretionary and financials. Interestingly, despite geopolitical risks, they do not significantly affect long-term volatility.

Our findings have two main implications. First, determining the direction and magnitude of the effect of macroeconomic factors on long-term stock volatility across sizes, styles, and sectors classification can improve the stability of financial markets and assist governments in responding to changes in macroeconomic factors. In the future, efforts to control housing prices in South Korea will be necessary because housing prices are the most dominant factor influencing stock volatility. In conclusion, a nuanced understanding of how different macroeconomic variables affect different segments of the stock market can lead to more targeted and effective investment strategies, especially in hybrid economies like South Korea. The findings of this study can guide investors in creating portfolios that are better aligned with macroeconomic trends and sector-specific dynamics. However, our study focuses on the impact of macroeconomic factors on the Korean stock market, which limits its applicability to other types of markets, such as developed or underdeveloped countries. In future research, the increasing influence of climate change policy uncertainty, a significant emerging factor in the global financial market, could benefit from including this aspect to better understand market volatility in the context of environmental policy changes.

## Author Contributions

**Conceptualization:** Chulyoung Cho, Jinseok Yang.

**Data curation:** Chulyoung Cho, Jinseok Yang.

**Formal analysis:** Chulyoung Cho, Jinseok Yang.

**Funding acquisition:** Beakcheol Jang.

**Investigation:** Chulyoung Cho, Jinseok Yang.

**Methodology:** Chulyoung Cho, Jinseok Yang.

**Project administration:** Beakcheol Jang.

**Resources:** Chulyoung Cho, Jinseok Yang.

**Software:** Chulyoung Cho, Jinseok Yang.

**Supervision:** Beakcheol Jang.

**Validation:** Chulyoung Cho, Jinseok Yang, Beakcheol Jang.

**Visualization:** Chulyoung Cho, Jinseok Yang.

**Writing – original draft:** Chulyoung Cho, Jinseok Yang.

**Writing – review & editing:** Chulyoung Cho, Jinseok Yang, Beakcheol Jang.

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
