## [Decision Letter · Decision Letter 0]

4 Jan 2024

PONE-D-23-34092Spectrum of Influence: Heterogeneous macroeconomic factors’ effects on stocks based on size, style, and sector in the South Korean marketPLOS ONE

Dear Dr. Jang,

Thank you for submitting your manuscript to PLOS ONE. After careful consideration, we feel that it has merit but does not fully meet PLOS ONE’s publication criteria as it currently stands. Therefore, we invite you to submit a revised version of the manuscript that addresses the points raised during the review process.

We look forward to receiving your revised manuscript.

Kind regards,

Ricky Chee Jiun Chia

Academic Editor

PLOS ONE

2. In the online submission form you indicate that your data is not available for proprietary reasons and have provided a contact point for accessing this data. Please note that your current contact point is a co-author on this manuscript. According to our Data Policy, the contact point must not be an author on the manuscript and must be an institutional contact, ideally not an individual. Please revise your data statement to a non-author institutional point of contact, such as a data access or ethics committee, and send this to us via return email. Please also include contact information for the third party organization, and please include the full citation of where the data can be found.

Reviewers' comments:

Reviewer's Responses to Questions

**Comments to the Author**

1. Is the manuscript technically sound, and do the data support the conclusions?

Reviewer #1: Partly

Reviewer #2: Yes

2. Has the statistical analysis been performed appropriately and rigorously? 

Reviewer #1: Yes

Reviewer #2: Yes

3. Have the authors made all data underlying the findings in their manuscript fully available?

Reviewer #1: Yes

Reviewer #2: Yes

4. Is the manuscript presented in an intelligible fashion and written in standard English?

Reviewer #1: Yes

Reviewer #2: No

5. Review Comments to the Author

Reviewer #1: Using the garch-midas method in association with other augmented tools, this paper evaluates low-frequency macro proxies’ effect on Korean stock market volatility. The authors claim that real estate prices play an important role in determining long-term stock volatility in Korea. I have the following concerns. First, the paper is unclear as to why the Korean market serves as a good laboratory for this particular study. Second, the paper looks more like an exercise without specifying the true marginal contribution to the extant and most recent literature that attempts to link macroeconomic and stock market performance via uncertainties. Third, the conclusion that housing prices are the main determinant of Korean stock volatility lacks theoretical background and economic influencing channels. Technical results sometimes may be biased for various reasons, which I suspect co-movement is the most convincing reason. Hence, it is natural to use this association to construct hedging or profitable strategies based on this conclusion. However, the paper spends less time in discussing investment implications, instead, the paper emphasizes a lot on prediction accuracy.

Reviewer #2: This study elucidates the macroeconomic variables that have a significant impact on stock volatility, using the Korean market as an example.The article also analyses the relationship between macroeconomic variables and stock return volatility indices.The research seems interesting, but the following questions still need to be further considered:

1. Formatting problems - Figure 1-8 is somewhat blurry and unclear, and the letters are not clearly visible.

2. Research paradigm - The article is very rich in research methods, including a series of models and formulas, can you write the background, advantages and limitations of the applied methods clearly, so as to facilitate the readers to read and study.

3. Introduction - The repetition of certain ideas or information within the text could be condensed to avoid redundancy. For example, the discussion about the impact of macroeconomic indicators on different stock sizes and sectors could be streamlined for clearer presentation.While the introduction states the significance of the study in analyzing the effects of macroeconomic variables on stock volatility, it could more explicitly define the specific research objectives and identify the existing gaps in the literature that this study intends to address. Clearly stating what this research aims to contribute to the existing knowledge base would enhance the paper's value.

4. Literature review - The literature review does not sufficiently highlight the relevance to the Korean stock market and may need to add more literature on the Korean stock market.

5. Research design - This study investigates the relationship between macroeconomic variables and stock volatility based on capital size, style, and sector in South Korea through an analysis of S&P South Korea BMI LargeMidCap, SmallCap, Growth, and Value, and Sectors Index values from January 2009 to September 2022. The article has no test for endogeneity of causality.This part of the article may need refining.

6. The article doesn't write about the limitations of the study.

7. English language logic and expression need to be improved

6. PLOS authors have the option to publish the peer review history of their article (what does this mean?). If published, this will include your full peer review and any attached files.

Reviewer #1: No

Reviewer #2: **Yes: **Xiansheng Chen

---

## [Author Response · Author response to Decision Letter 0]

11 Feb 2024

PONE-D-23-34092 - Reviewer 1

Heterogeneous macroeconomic factors’ effects on stocks across sizes, styles, and sectors in the South Korean market

Thank you for another opportunity to revise our manuscript. Our revision again benefitted from your and the editor’s writing guidance. We made a sincere effort to address all of the comments we received and, as a result, have made substantial writing changes to the paper. We hope that these changes and clarifications significantly improve the paper and alleviate the concerns you and the reviewer raised about the previous draft. We look forward to your feedback and would, of course, be happy to make further changes.

Below we provide responses to your comments on our paper (reproduced in bold, blue italics).

Comment 1: I have the following concerns. First, the paper is unclear as to why the Korean market serves as a good laboratory for this particular study.

Author Response: Thank you for pointing this out. We’ve clearly explained why the Korea market is a good laboratory for this study. 

[Line 18 (p.2) ~ Line 38 (p.3)]:

 “Over the past two decades, the global economy has faced several tumultuous events, such as the 2008 financial crisis, the 2010 Eurozone Debt Crisis, and the COVID-19 pandemic. These events triggered profound macroeconomic shocks, exemplified by significant fluctuations in the stock market [1, 2]. The aftermath of these crises has spotlighted the pivotal role of macroeconomic indicators—such as interest rates and money supply—in influencing market dynamics, a relationship that has become even more pronounced in the post-pandemic world [3]. Evidence increasingly suggests a tangible link between these indicators and stock market fluctuations, pointing towards a complex interplay that demands deeper investigation [4-6]. However, the focus of existing research has largely been confined to the developed markets, with a significant emphasis on the US stock market. This narrow lens raises questions about the universality of these findings, underscoring a critical gap in our understanding of how different market contexts might influence or alter these established dynamics. In this context, the South Korean market emerges as a particularly compelling subject for in-depth study, given its unique position at the nexus of developed and emerging market characteristics. South Korea's economy and stock market have shown distinctive responses to these global macroeconomic shocks, unlike the often-studied like the US [7]. There are limited studies that focus on economies that combine the characteristics of both emerging and developed markets. This study attempts to fill this gap by examining the South Korean stock market, a unique blend of emerging and developed market characteristics [8]. The uniqueness is rooted in South Korea's rapid industrialization, technological innovation, and the central role in exports, alongside geopolitical tensions and global economic dependencies [9, 10]. Such a complex economic background makes South Korea an ideal case study for investigating the nuanced effects of macroeconomic variables on stock market dynamics.”

Comment 2: Second, the paper looks more like an exercise without specifying the true marginal contribution to the extant and most recent literature that attempts to link macroeconomic and stock market performance via uncertainties. 

Author Response: Thank you very much for this suggestion! We agree that it would be good for us to clearly describe the marginal contribution to the extant and most recent literature, and we have revised what this research aims to contribute:

[Line 98 (p.5) ~ Line 114 (p.5)]:

“Our study contributes threefold to the existing literature. First, we examine the diverse effects across all sectors, capital sizes, and investment styles (growth and value), going beyond traditional analyses that focus on macroeconomic indicators and their impact on national indices or specific sectors. It provides a comprehensive and nuanced cross-sectional analysis that enriches our understanding beyond the current literature.

Second, we build on this foundation by expanding the analytical framework to include a broader range of variables that affect sectors, sizes, and styles. Traditional research is often limited to traditional macroeconomic variables or alternative economic indicators. By integrating a variety of indicators, such as the category-specific EPU, GPRI and EMV, along with the Business Sentiment Index (BSI) and the Economic Sentiment Index (ESI), we provide an updated and detailed view of investor behavior and market conditions. This enhanced approach significantly deepens traditional economic analysis.

Third, this study, which focuses on the South Korean stock market, provides in-depth insights and a sophisticated understanding of countries that have characteristics of both developed and emerging markets and are critical to the global economy. This analysis helps investors develop better-informed strategies while expanding our understanding of the dynamics of such hybrid markets. For investors trying to make sense of these complicated market conditions, the knowledge gained from this study is invaluable. In addition, we evaluate the effectiveness of key variables during recent economic challenges, including the post-COVID-19 downturn, the devaluation of the Korean won, escalating inflation, and rising interest rates. We test the predictive power of our selected variables during a period of significant stock market volatility since the second quarter of 2020. The results demonstrate the robustness of our methodology and provide valuable insights into the current financial market.”

Comment 3: Third, the conclusion that housing prices are the main determinant of Korean stock volatility lacks theoretical background and economic influencing channels. 

Author Response: Thank you for this suggestion! We've added the suggested content to explain the theoretical background and empirical research of the housing price impacts on the stock markets:

[Line 474 (p.24) ~ Line 479 (p.24)]:

“The housing prices significantly influence stock market volatility in Korea is underpinned by the wealth effect theory and the credit channel theory. The wealth effect posits that changes in housing prices impact consumer wealth and spending, which in turn affects stock markets [53]. In addition, the credit channel theory suggests that housing market values affect borrowing capacity, which in turn affects investment behaviors in the context of stock volatility [54]. Empirical studies, such as Kim work on the Korean market [55]. Our results are consistent with the above theoretical framework and Kim’s empirical results on the Korean market, confirming the link between theory and practice.”

Comment 4: Technical results sometimes may be biased for various reasons, which I suspect co-movement is the most convincing reason. Hence, it is natural to use this association to construct hedging or profitable strategies based on this conclusion. However, the paper spends less time in discussing investment implications, instead, the paper emphasizes a lot on prediction accuracy.

Author Response: Thank you for this suggestion! We believe that it would be great to demonstrate the investment implications. We’ve added the suggested content in the conclusion.

[Line 462 (p.23) ~ Line 466 (p.23)]:

“This suggests investors might consider adjusting their investment strategies across stock sizes during different economic cycles. For instance, a focus on SmallCap stocks could be beneficial during periods of monetary expansion. The significant impact of the housing price-sales index on sector volatility, particularly in real estate-heavy sectors, suggests that sector-specific strategies are critical. Investors should monitor housing market trends for potential shifts in sector performance.”

[Line 470 (p.23) ~ Line 472 (p.23)]:

“This suggests that the sensitivity of growth stocks to uncertainty and value stocks to business conditions provides a strategic insight for investors considering style-based investing. The diversification of portfolios to include a mix of growth and value stocks can potentially be a hedge against market volatility due to macroeconomic changes.”

[Line 489 (p.25) ~ Line 492 (p.25)]:

“In conclusion, a nuanced understanding of how different macroeconomic variables affect different segments of the stock market can lead to more targeted and effective investment strategies, especially in hybrid economies like South Korea. The findings of this study can guide investors in creating portfolios that are better aligned with macroeconomic trends and sector-specific dynamics.”

PONE-D-23-34092 - Reviewer 2

Heterogeneous macroeconomic factors’ effects on stocks across sizes, styles, and sectors in the South Korean market

Thank you for another opportunity to revise our manuscript. Our revision again benefitted from your and the editor’s writing guidance. We made a sincere effort to address all of the comments we received and, as a result, have made substantial writing changes to the paper. We hope that these changes and clarifications significantly improve the paper and alleviate the concerns you and the reviewer raised about the previous draft. We look forward to your feedback and would, of course, be happy to make further changes.

Below we provide responses to your comments on our paper (reproduced in bold, blue italics).

Comment 1: Formatting problems - Figure 1-8 is somewhat blurry and unclear, and the letters are not clearly visible.

Author Response: Thank you for pointing out this. We have updated figures to the high resolutions so that the letters are clearly visible.

Comment 2: Research paradigm - The article is very rich in research methods, including a series of models and formulas, can you write the background, advantages and limitations of the applied methods clearly, so as to facilitate the readers to read and study.

Author Response: We think this is an excellent suggestion. We have added the background, advantages and limitations. This revised text reads as follows on [Line 87 (p.4) ~ Line 95 (p.5)]:

“The use of macroeconomic factors presents a data frequency issue, as these variables often have low frequencies, while stock market volatility is high. The mixed data sampling(MIDAS) method, which differentiates between short- and long-term stock market volatility, addresses this by incorporating low-frequency variables into the long-term component, solving the data cycle problem without information loss. We employ multivariate Generalized Autoregressive Conditional Heteroskedasticity (GARCH) MIDAS model combined with the adaptive lasso technique to find the utmost impact macroeconomic factors on stock markets [24-29]. As to the inherent limitations, we are not able to completely remove the potential endogeneity issue from omitted variables. However, since we include the most of the well-known macroeconomic variables as control, the model can minimize the endogeneity issue as much as possible.”

Comment 3: Introduction - The repetition of certain ideas or information within the text could be condensed to avoid redundancy. For example, the discussion about the impact of macroeconomic indicators on different stock sizes and sectors could be streamlined for clearer presentation.

Author Response: We appreciate the reviewer's feedback. We've removed and streamlined to avoid redundancy.

[Line 39 (p.3) ~ Line 50 (p.3)]: Removed

[Line 65 (p.3) ~ Line 72 (p.4)]: Removed

While the introduction states the significance of the study in analyzing the effects of macroeconomic variables on stock volatility, it could more explicitly define the specific research objectives and identify the existing gaps in the literature that this study intends to address. 

Author Response: We appreciate the reviewer's feedback. We've removed and added suggested contents.

[Line 34 (p.3) ~ Line 38 (p.3)] – More explicitly define the specific research objectives.

 “The main objective of this study is to investigate which factor – among traditional macroeconomic indicators and alternative economic indicators – exerts the most significant influence on stock market volatility. This includes examining how these factors influence stock volatility across different sectors, capital sizes, and investment styles (growth or value). Thereby, this study enriches the understanding of hybrid market dynamics and provide extended insights for volatility, offering valuable guidance to investors and policymakers.”

Clearly stating what this research aims to contribute to the existing knowledge base would enhance the paper's value.

Author Response: We appreciate the reviewer's feedback. We've added the true marginal contributions to enhance the paper's value.

[Line 98 (p.5) ~ Line 114 (p.5)]– Clearly Describe the contributions of this study

“Our study contributes threefold to the existing literature. First, we examine the diverse effects across all sectors, capital sizes, and investment styles (growth and value), going beyond traditional analyses that focus on macroeconomic indicators and their impact on national indices or specific sectors. It provides a comprehensive and nuanced cross-sectional analysis that enriches our understanding beyond the current literature.

Second, we build on this foundation by expanding the analytical framework to include a broader range of variables that affect sectors, sizes, and styles. Traditional research is often limited to traditional macroeconomic variables or alternative economic indicators. By integrating a variety of indicators, such as the category-specific EPU, GPRI and EMV, along with the Business Sentiment Index (BSI) and the Economic Sentiment Index (ESI), we provide an updated and detailed view of investor behavior and market conditions. This enhanced approach significantly deepens traditional economic analysis.

Third, this study, which focuses on the South Korean stock market, provides in-depth insights and a sophisticated understanding of countries that have characteristics of both developed and emerging markets and are critical to the global economy. This analysis helps investors develop better-informed strategies while expanding our understanding of the dynamics of such hybrid markets. For investors trying to make sense of these complicated market conditions, the knowledge gained from this study is invaluable. In addition, we evaluate the effectiveness of key variables during recent economic challenges, including the post COVID-19 downturn, the devaluation of the Korean won, escalating inflation, and rising interest rates. We test the predictive power of our selected variables during a period of significant stock market volatility since the second quarter of 2020. The results demonstrate the robustness of our methodology and provide valuable insights into the current financial market.”

Comment 4: . Literature review - The literature review does not sufficiently highlight the relevance to the Korean stock market and may need to add more literature on the Korean stock market.

Author Response: Thank you pointing this out. The reviewer is correct, and we have added more literature and the reason to choose the Korean market as a good laboratory. This revised text reads as follows on [Line 18 (p.2) ~ Line 34 (p.3)]:

“Over the past two decades, the global economy has faced several tumultuous events, such as the 2008 financial crisis, the 2010 Eurozone Debt Crisis, and the COVID-19 pandemic. These events triggered profound macroeconomic shocks, exemplified by significant fluctuations in the stock market [1, 2]. The aftermath of these crises has spotlighted the pivotal role of macroeconomic indicators—such as interest rates and money supply—in influencing market dynamics, a relationship that has become even more pronounced in the post-pandemic world [3]. Evidence increasingly suggests a tangible link between these indicators and stock market fluctuations, pointing towards a complex interplay that demands deeper investigation [4-6]. However, the focus of existing research has largely been confined to the developed markets, with a significant emphasis on the US stock market. This narrow lens raises questions about the universality of these findings, underscoring a critical gap in our understanding of how different market contexts might influence or alter these established dynamics. In this context, the South Korean market emerges as a particularly compelling subject for in-depth study, given its unique position at the nexus of developed and emerging market characteristics. South Korea's economy and stock market have shown distinctive responses to these global macroeconomic shocks, unlike the often-studied like the US [7]. There are limited studies that focus on economies that combine the characteristics of both emerging and developed markets. This study attempts to fill this gap by examining the South Korean stock market, a unique blend of emerging and developed market characteristics [8]. The uniqueness is rooted in South Korea's rapid industrialization, technological innovation, and the central role in exports, alongside geopolitical tensions and global economic dependencies [9, 10]. Such a complex economic background makes South Korea an ideal case study for investigating the nuanced effects of macroeconomic variables on stock market dynamics”

Comment 5: Research design - This study investigates the relationship between macroeconomic variables and stock volatility based on capital size, style, and sector in South Korea through an analysis of S&P South Korea BMI LargeMidCap, SmallCap, Growth, and Value, and Sectors Index values from January 2009 to September 2022. The article has no test for endogeneity of causality. This part of the article may need refining.

Author Response: Thank you for pointing out this. We’ve demonstrated the possible endogeneity due to omitted variables but rendering the study’s findings are still valid and meaningful despite of the acknowledged limitations. 

[Line 93 (p.5) ~ Line 95 (p.5)]:

“As to the inherent limitations, we are not able to completely remove the potential endogeneity issue from omitted variables. However, since we include the most of the well-known macroeconomic variables as control, the model can minimize the endogeneity issue as much as possible.”

Comment 6: The article doesn't write about the limitations of the study.

Author Response: Thank you very much for this suggestion. It would be great to add the limitations of the study and the future work.

[Line 492 (p. 25) ~ Line 496 (p.25)]: 

“However, our study focuses on the impact of macroeconomic factors on the Korean stock market, which limits its applicability to other types of markets, such as developed or underdeveloped countries. In future research, the increasing influence of climate change policy uncertainty, a significant emerging factor in the global financial market, could benefit from including this aspect to better understand market volatility in the context of environmental policy changes.”

Comment 7: English language logic and expression need to be improved

Author Response: Thank you for pointing this out. We’ve been corrected all spelling and grammatical errors. Moreover, we have been revised the manuscript based on the manuscript body formatting guidelines.

---

## [Editor Report · Decision Letter 1]

27 Feb 2024

Heterogeneous macroeconomic factors’ effects on stocks across sizes, styles, and sectors in the South Korean market

PONE-D-23-34092R1

Dear Dr. Beakcheol Jang,

We’re pleased to inform you that your manuscript has been judged scientifically suitable for publication and will be formally accepted for publication once it meets all outstanding technical requirements.

Kind regards,

Ricky Chee Jiun Chia

Academic Editor

PLOS ONE
---

## [Editor Report · Acceptance letter]

5 Apr 2024

PONE-D-23-34092R1 

PLOS ONE

Dear Dr. Jang, 

I'm pleased to inform you that your manuscript has been deemed suitable for publication in PLOS ONE. Congratulations! Your manuscript is now being handed over to our production team.

Kind regards, 

on behalf of

Dr. Ricky Chee Jiun Chia 

Academic Editor

PLOS ONE